# Optical manipulation of single flux quanta

I.S. Veshchunov[1,2,3,*], W. Magrini[1,2,4,*], S.V. Mironov[3,4], A.G. Godin[1,2], J.-B. Trebbia[1,2], A.I. Buzdin[4], Ph. Tamarat[1,2] & B. Lounis[1,2]

Magnetic field can penetrate into type II superconductors in the form of Abrikosov vortices, which are magnetic flux tubes surrounded by circulating supercurrents often trapped at defects referred to as pinning sites. Although the average properties of the vortex matter in superconductors can be tuned with magnetic fields, temperature or electric currents, handling of individual Abrikosov vortices remains challenging and has been demonstrated only with sophisticated scanning local probe microscopies. Here we introduce a far-field optical method based on local heating of the superconductor with a focused laser beam to realize a fast and precise manipulation of individual vortices, in the same way as with optical tweezers. This simple approach provides the perfect basis for sculpting the magnetic flux profile in superconducting devices like a vortex lens or a vortex cleaner, without resorting to static pinning or ratchet effects.

[1] Université de Bordeaux, LP2N, F-33405 Talence, France. [2] Institut d'Optique & CNRS, LP2N, F-33405 Talence, France. [3] Moscow Institute of Physics and Technology, 141700 Dolgoprudny, Russia. [4] Université de Bordeaux, LOMA, F-33405 Talence, France. * These authors contributed equally to this work. Correspondence and requests for materials should be addressed to B.L. (email: brahim.lounis@u-bordeaux.fr).

Early experiments showed that the average properties of the vortex matter can be tuned with magnetic fields[1], temperature or electric currents[2]. A first step towards the controlled manipulation of vortices has been achieved by moving magnetic Bloch walls of a ferrite garnet film used to image single vortex locations[3]. Yet, handling of individual vortices has been performed only with magnetic force[4,5], superconducting quantum interference device[6–8] or strain-induced[9] scanning local probe microscopies. Since these techniques are slow and heavy to implement in cryogenic environments, new approaches to provide a large-scale and versatile basis for sculpting the magnetic flux profile in superconductor devices are required. Here we introduce a far-field optical method based on local heating of the superconductor with a focused laser beam to realize a fast and precise manipulation of individual Abrikosov vortices. Since a single vortex can induce a Josephson phase shift[10,11], our method paves the way to fast optical drive of Josephson junctions, notably in superconducting elementary circuits with potential large parallelization of operations.

Although the possibility to induce a global vortex flow by thermal gradients in superconductors (SCs) was experimentally demonstrated almost 50 years ago[12,13], thermal manipulation of individual Abrikosov vortices has not yet been achieved. The early observations of vortex flows in weak thermal gradients were interpreted on the basis that vortices behave as entropy-carrying particles, which obey general thermal diffusion laws and therefore are in search of colder places in the superconductor. However, theoretical investigations revisiting the problem of entropy transport in superconductors show that the vortex superconducting current does not carry entropy[14]. In the limit of a London magnetic penetration depth $\lambda$ much larger than the coherence length $\xi$, the vortex core is very small and its contribution to the energy can be neglected[15]. The free energy per unit length of an isolated vortex is thus proportional to the density of Cooper pairs and is given by

$$U \approx \frac{\Phi_0^2}{4\pi\mu_0\lambda^2}\ln\left(\frac{\lambda}{\xi}\right), \qquad (1)$$

where $\Phi_0$ is the flux quantum and $\mu_0$ the vacuum permeability. It linearly grows as $T_c - T$ when reducing the temperature $T$ down from the superconducting critical temperature $T_c$ (refs 15,16). As a consequence, a temperature gradient in the superconductor will generate a thermal force per vortex unit length given by

$$\mathbf{F} \approx \frac{\Phi_0^2}{4\pi\mu_0\lambda_0^2}\ln\left(\frac{\lambda_0}{\xi_0}\right)\frac{\nabla T}{T_c}, \qquad (2)$$

where $\lambda_0$ and $\xi_0$ are the values of $\lambda$ and $\xi$ at zero temperature. This force will therefore drive vortices towards higher temperature regions.

In this work, we show that a tightly focused laser beam inducing a strong thermal gradient can be used to manipulate single flux quanta. The laser locally heats the superconductor and creates a micron-sized hotspot with a temperature rise in the Kelvin range, while keeping the temperature below $T_c$. The large thermal gradient can easily be tuned with laser power, so that the generated thermal force overcomes the pinning potential and induces a vortex motion towards the laser focus. Therefore, the laser beam acts as optical tweezers that move single flux quanta to any new desired positions in the superconductor.

## Results

**Vortex manipulation.** The experimental set-up used for single vortex optical manipulation and magneto-optical imaging is depicted in Fig. 1a. A 90 nm-thick niobium film was cooled below its critical temperature $T_c = 8.6\,\mathrm{K}$ under a weak external magnetic field $H_{\mathrm{ext}}$ applied perpendicular to the film, to set the SC in the mixed state. Figure 1b is a magneto-optical image of a spontaneous vortex distribution obtained at $T = 4.6\,\mathrm{K}$ and for $H_{\mathrm{ext}} = 0.024\,\mathrm{Oe}$. The vortices are located at pinning sites that are randomly distributed in the SC sample. Vortex after vortex, they are then optically dragged and repositioned into an artificial pattern forming the letters A V for Abrikosov vortices, as shown in Fig. 1c (see also the vortex rearrangement movie in the Supplementary Movie 1).

Because they dissipate energy and generate internal noise, vortices constitute a serious obstacle limiting the operation of numerous superconducting devices[17]. The most desirable method to overcome this difficulty would be to remove the vortices from the bulk of the superconductor without the need for material structuration or the incorporation of impurities and defects[18,19]. To this purpose, we show in Fig. 1d,e how a vortex-free area is produced in a niobium film by simply scanning a focused laser beam, which picks up the vortices in its path, drags and drops them in a bordering area of the superconductor, like a vortex broom. Actually, because of their mutual repulsion, all vortices cannot be piled up in the same optical spot. This explains the comet-like shape of the vortex distribution in the region along the laser path, with a maximal vortex density at the final position of the laser spot (see the vortex-cleaning movie in Supplementary Movie 2).

To perform an efficient single vortex manipulation, it is crucial to choose a laser power low enough to keep the local temperature below $T_c$ and high enough so that the thermal force overcomes the pinning potentials. To determine the absorbed power needed to untrap all single vortices in a selected region of the SC at temperature $T$, we sequentially position the laser spot at a micrometric distance ($\sim 1\,\mu\mathrm{m}$) from each vortex (crosses in Fig. 2a) and count the number of untrapped vortices for various laser powers. As displayed in Fig. 2b, the fraction of untrapped vortices strongly depends on the laser power and the base temperature, reflecting a large distribution of pinning potentials in the sample. Most notably, we demonstrate in Fig. 2c that 100% of the vortices in the selected region can be dragged out of their pinning sites. Moreover, we checked that the superconductivity is not destroyed at the laser powers used for the manipulation. For this, we have compared the magneto-optical images of vortices under laser illumination with that of vortices at different SC temperatures without laser (Supplementary Figs 1 and 2, and Supplementary Note 1). Since the vortex image profile (width and contrast) is slightly affected by the laser, we conclude that the local temperature remains well below $T_c$ when performing the vortex manipulation.

This approach allows a straightforward estimation of the pinning force of each vortex at any temperature. From Fig. 2b, we can estimate the pinning force of the most strongly bound vortices in the selected SC region at various temperatures. At $T = 4.6\,\mathrm{K}$ for instance, the thermal force overcomes all pinning forces for an absorbed power of $13\,\mu\mathrm{W}$, corresponding to a temperature gradient at the vortex position of $\sim 1.4\,\mathrm{K}\,\mu\mathrm{m}^{-1}$ (Supplementary Fig. 3 and Supplementary Note 2). Using expression (2) of the thermal force and taking $\lambda_0 = 90\,\mathrm{nm}$ and $\lambda_0/\xi_0 = 9$ relevant to our niobium film, we deduce a maximal pinning force per vortex unit length $F_{\mathrm{p}} \sim 14\,\mathrm{pN}\,\mu\mathrm{m}^{-1}$. This order of magnitude estimate is in accordance with previous determinations based on critical current measurements on similar samples[20–23]. Finally, the temperature dependence of the pinning force of a strongly bound vortex is displayed in Fig. 2d. It is well reproduced with the empirical power law $F_{\mathrm{p}} \propto (1 - T/T_c)^\gamma$ with $\gamma = 3.4$, in agreement with previous ensemble measurements on vortices in niobium films[20,24].

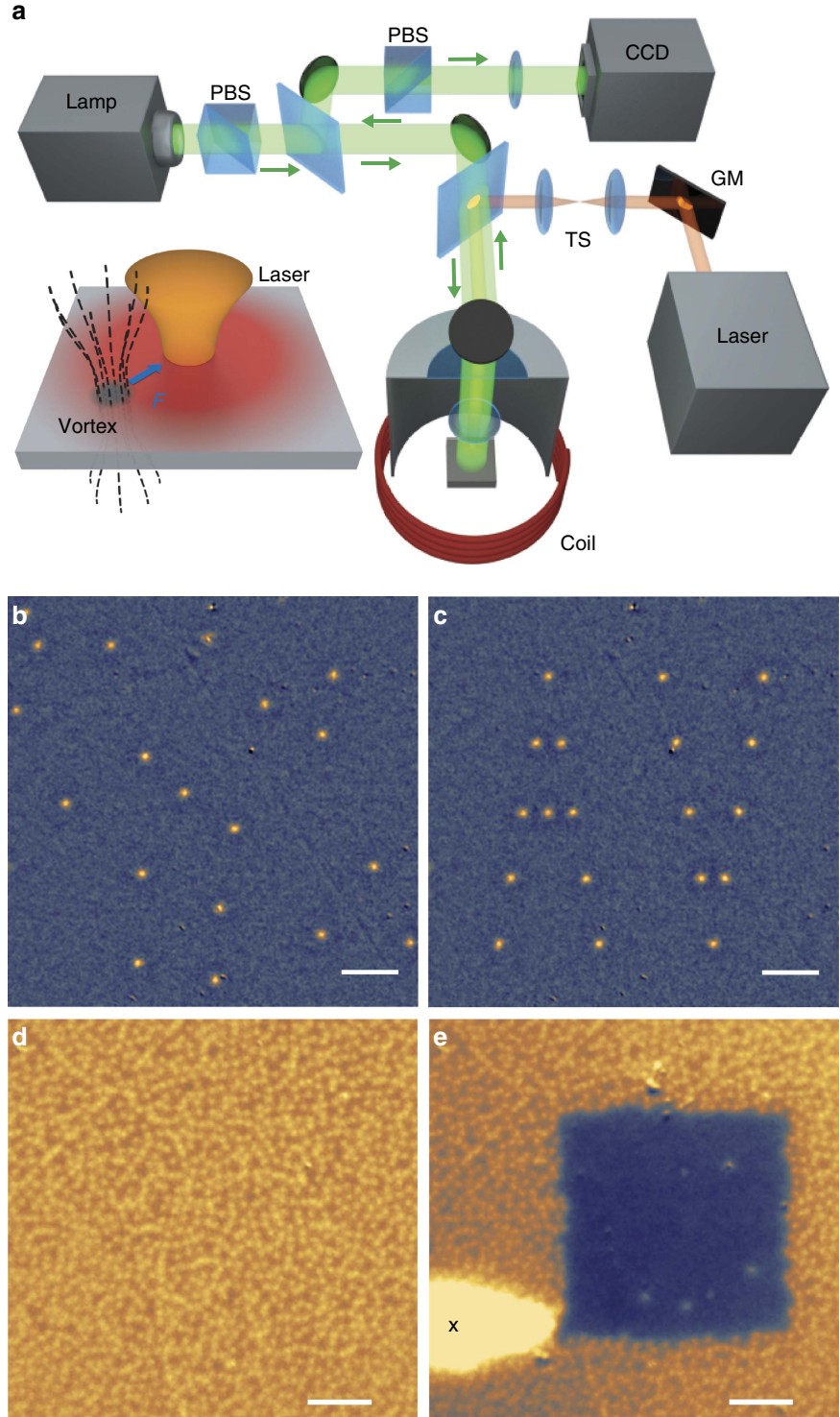

**Figure 1 | Single vortex manipulation with a focused laser beam.** (**a**) The concept of vortex attraction in a thermal gradient induced by a laser spot is illustrated. Magneto-optical imaging of individual vortices is based on the Faraday rotation of light polarization in a Bi:LuIG garnet layer placed onto the superconductor, in a crossed-polarizer beam path configuration[30-32]. PBS, polarizing beam-splitter. CCD, charge-coupled device. Local heating of the niobium film is performed with a tightly focused continuous wave laser (wavelength 561 nm) from which 40% of the optical power is absorbed. Vortex manipulation is performed by moving the laser beam with galvanometric mirrors (GMs) placed in a telecentric system (TS). (**b**) Magneto-optical image of a field-cooled vortex structure in the niobium film under $H_{ext} = 0.024$ Oe at $T = 4.6$ K. (**c**) Artificial vortex pattern engineered by single vortex repositioning from the initial vortex distribution displayed in **b**. The repositioning procedure is fully automatized, as described in the Methods section. The laser is focused on the SC with a full-width at half-maximum diameter of 1.1 μm. The absorbed power is set to 17 μW. (**d**) A new spontaneous vortex distribution is generated after a thermal cycle above $T_c$ and field cooling back to $T = 4.6$ K under $H_{ext} = 1.64$ Oe. (**e**) From the initial vortex distribution displayed in **d**, a vortex-free area is produced by scanning the laser spot like a vortex broom. The ending point of the spot trajectory is marked with a black cross. All scale bars are 20 μm.

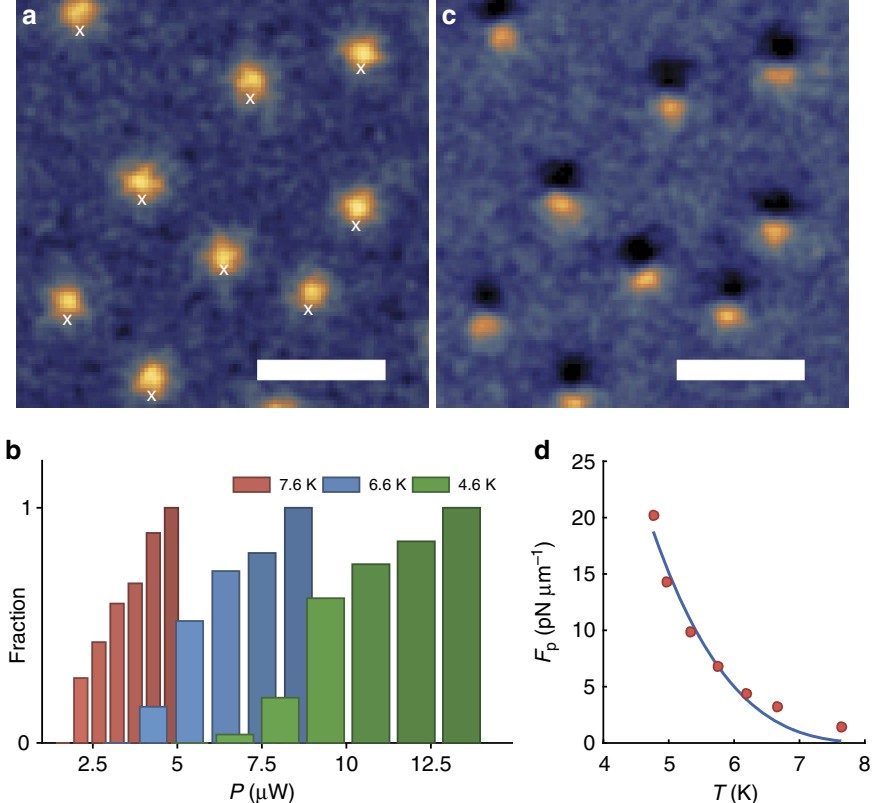

**Figure 2 | Laser power and temperature effects on vortex manipulation.** (**a**) Initial magneto-optical image of an area of the niobium film cooled at $T = 4.6$ K in a magnetic field $H_{ext} = 0.22$ Oe. A laser spot is then successively placed at a fixed distance of 1.1 µm from each vortex. The central positions of the laser spot are marked with white crosses. (**b**) Histograms of the fraction of untrapped vortices as a function of the absorbed laser power, for three different base temperatures of the SC. The statistics are built from 30 vortices. (**c**) Image of the same area, built from the difference between magneto-optical-imaging contrasts after and before laser heating with an absorbed power of 13 µW. In these conditions, all nine vortices have moved. (**d**) Temperature dependence of the pinning force of a strongly bound vortex. The solid curve is a fit with the empirical power law $F_{p} \propto (1 - T/T_c)^{\gamma}$, yielding the exponent $\gamma = 3.4$. All scale bars are 10 µm.

**Shaping magnetic flux in a superconductor with light**. The extreme simplicity of optical generation of strong local thermal gradients in SCs enables to tailor normal metal regions in the SC, and study the magnetic flux penetration close to their boundaries. For instance, starting from a spontaneous spatial distribution of vortices, a strong illumination of the Nb film with a focused laser produces a central region of radius $R_0$, where the temperature exceeds $T_c$. This normal (N) region thus traps a magnetic flux $\Phi \sim \pi H_{ext} R_0^2$ originating from the destroyed vortices (Fig. 3a). The averaged radial profile of the magnetic field during laser illumination is presented in Fig. 3b. It recalls that of a supercurrent loop of radius $R_0$ surrounding the N region (dashed blue curve in Fig. 3b), which adds to the average magnetic field in the superconductor (defined as the total flux through the SC divided by the SC surface). After switching the laser off, the local temperature starts relaxing to its base value, inducing shrinkage of the N region, where $T$ remains larger than $T_c$. The magnetic flux remains trapped within the N region as a result of the geometrical barrier[25,26] that prevents the vortices from entering the superconducting region of the sample. To support this picture, we solve the heat equation to calculate the time evolution of the sample temperature profile at early times after the laser switch-off (red curves of Fig. 3e). For each profile, the solution of the stationary Ginzburg–Landau equation gives the corresponding spatial distribution of the normalized order parameter (blue curves) and therefore the size of the shrinking N region (Supplementary Fig. 4 and Supplementary Note 3). Because of flux conservation, the trapped magnetic field increases

with time (green levels) until it reaches the geometrical barrier critical field (penetration field $H_p$) for a radius $R^* = R_0 \sqrt{H_{ext}/H_p}$. When the radius of the N region becomes smaller than $R^*$, the geometrical barrier vanishes and the vortices penetrate the SC region where they get trapped at the nearest pinning sites. This behaviour explains the final distribution of flux quanta in Fig. 3c, characterized by a dense vortex region with radius $R^*$ (where the individual vortices are not optically resolved) belt by a vortex-free SC region with an external radius $R_0$. Following the Bean critical state model[16,27], the vortex distribution is determined by the balance between the pinning force and the Lorentz driving force $j_c \Phi_0$, $j_c$ being the critical current density. This explains why the radial profile of the magnetic field presented in Fig. 3d is almost linear in the vortex cluster (Supplementary Note 4). Interestingly, the experimental dependence of the ratio $R^*/R_0$ on the external magnetic field $H_{ext}$ plotted in Fig. 3f is very well described by the theory of the geometrical barrier (Supplementary Note 5 for details).

## Discussion

The performed experiments unveil new aspects of the interaction between laser radiation and the vortex matter. Since the concept of attractive thermal force exerted on an isolated vortex is new, further theoretical investigations are required to gain a deeper understanding of all contributions of this force, including the contribution of the vortex core and of the supercurrents around the core.

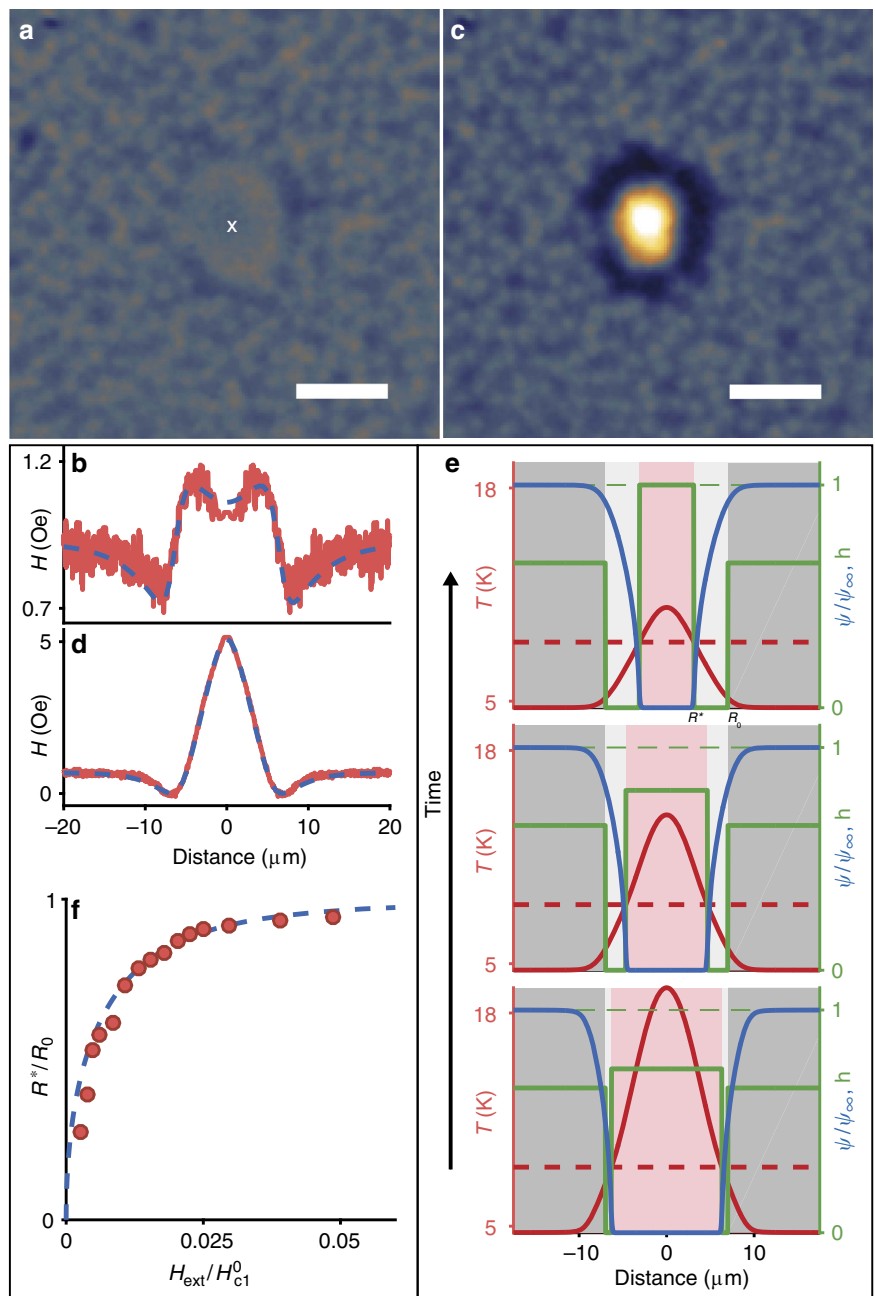

**Figure 3 | Dynamics of the magnetic flux penetration through a moving SC/N boundary.** (**a**) Magnetic field landscape during laser heating, for $H_{ext} = 3.0$ Oe and a base SC temperature $T = 4.6$ K. The central position of the laser spot is marked with a white cross. The absorbed power 450 µW sets a central region with radius $R_0 = 6.5$ µm in a N state. (**b**) Averaged radial profile of the magnetic field. The theoretical profile (blue dashed curve) is obtained from the magnetic field created by a supercurrent loop surrounding the N region. (**c**) After the laser switch-off, a dense vortex cluster with radius $R^\star = 5.5$ µm is surrounded by a vortex-free area with external radius $R_0$. (**d**) Averaged radial profile of the magnetic field. The theoretical profile (blue dashed curve) is obtained from the Bean critical state model. (**e**) Model of temporal evolution of the magnetic field penetration through the SC region after the laser switch-off. While the temperature profile (red curve) collapses, the profile of the order parameter (blue curve) tightens. The magnetic field h (normalized to the barrier penetration field) trapped in the N region increases (green levels) and will penetrate the SC region only when exceeding the geometrical barrier, that is, for a N-region radius smaller than $R^\star$. (**f**) Dependence of the ratio $R^\star/R_0$ on the external magnetic field $H_{ext}/H_{c1}^0$, where $H_{c1}^0$ is the critical field at zero temperature. The blue curve is deduced from a theoretical model of geometrical barrier developed in the Supplementary Note 4. All scale bars are 10 µm.

By choosing the appropriate laser parameters, we showed that one can perform a safe and efficient manipulation of single vortices, with a 100% rate of success. In practice, the distance over which a vortex can be dragged is only limited by the field of view of the microscope objective, which is of the order of a millimetre

in our case. We realized various regimes of vortex manipulation, from the precise and rapid positioning of individual vortices to the generation of tight vortex bunches.

In the present work we implemented a proof-of-principle vortex broom and aimed at recording the vortex broom at work,

using our magneto-optical imaging system. In Supplemental Movie 2, we chose a low manipulation speed to image the movements of vortices with a sufficient contrast. The complete cleaning displayed in this movie took 420 s, corresponding to an average vortex speed of 10 μm s$^{-1}$. Actually, the laser scanning system, including galvanometric mirrors and the driving electronics, limits the cleaning speed in our set-up. An area of $70 \times 70$ μm could be cleaned in 350 ms, corresponding to $\sim 6$ mm s$^{-1}$ for the vortex drive speed. Faster scanners would increase the vortex driving speed, up to a maximal speed which should be given by the ratio of the hotspot size ($\sim 1$ μm) to the thermal response time of the sample ($\sim 1$ ns), of the order of 1 km s$^{-1}$.

We foresee that the versatility of our optical method of vortex manipulation will fuel fundamental investigations of the vortex matter and vortex dynamics in Abrikosov lattices. Moreover, the interplay between photons and single flux quanta should open up novel research directions in quantum computation based on braiding and entanglement of vortices, Josephson switches of electric current[28], or optically controlled elements of Rapid Single Flux Quantum logics[29]—a new research field that could be called optofluxonics.

## Methods

**Magneto-optical imaging of vortices in a niobium film.** Magneto-optical imaging of individual vortices is based on the Faraday rotation of light polarization in a magneto-optical indicator placed onto the superconductor, in a crossed-polarizer beam path configuration. The superconductor is a niobium film of thickness 90 nm grown by magnetron sputtering on a silicon substrate (thickness 150 μm). The indicator is a 2.5 μm-thick Bi:LuIG garnet of composition $Lu_{3-x}Bi_xFe_{5-z}Ga_zO_{12}$ with $x \sim 0.9$ and $z \sim 1.0$, with in-plane uniform saturation magnetization 50 G, which was grown by liquid phase epitaxy on a (100) oriented paramagnetic gadolinium–gallium-garnet $Gd_3Ga_5O_{12}$ substrate. This garnet has a high Verdet constant of $\sim 0.06°$ μm$^{-1}$ mT$^{-1}$ at light wavelengths around 560 nm.

The sample was mounted on a piezo scanner and inserted together with an aspheric lens (numerical aperture 0.49) in a closed-cycle helium cryostat. It was then submitted to a perpendicular external magnetic field $H_{ext}$ and cooled below the critical temperature $T_c = 8.6$ K to set the niobium film in the SC mixed state.

The magneto-optical contrast strongly depends on the extinction ratio of the ensemble polarizer-lens-crossed analyser[30,31]. An extinction ratio $\sim 10^{-3}$ at cryogenic temperatures could be achieved. To enhance the contrast of magneto-optical imaging, a background subtraction procedure was used to suppress the contribution of defects at the sample surface and non-uniformity of the sample illumination. An image taken at $T > T_c$ under external magnetic field was subtracted from all raw images recorded at $T < T_c$ under the same applied magnetic field.

**Creating artificial vortex patterns.** Software was developed to locate all vortices of a selected area in a magneto-optical image and to relocate them to the new desired positions. An important point is to make sure that the laser beam manipulating a chosen vortex does not displace any other one during the whole vortex trajectory. The positions of the vortex centres are initially identified with centroid calculations and used as seeds of a Voronoi diagram. The trajectory repositioning a single vortex from its initial location to its final position is then generated along segments of the Voronoi diagram so that the distance between the dragged vortex and the fixed vortices is maximized along the path.

**Order parameter profile across the N/SC boundary.** To study the evolution of vortex distribution after heating the SC above $T_c$ with strong illumination we solve the Ginzburg–Landau equation to derive the spatial distribution of the order parameter for each temperature profile during the thermal relaxation.

Since the relaxation time of the order parameter $\psi$ is much smaller than that of the temperature profile, we consider an adiabatic evolution of $\psi$, i.e., $\partial_t\psi = 0$. Moreover, considering that the scale of the order parameter variation is much smaller than the hot-spot characteristic size, the term with the first derivative $\partial_r\psi$ can be neglected in $\Delta\psi$, so that the Ginzburg–Landau equation writes

$$-\frac{\hbar^2}{4m}\frac{\partial^2\psi}{\partial r^2} + \alpha[T(r) - T_c]\psi + \beta\psi^3 = 0. \qquad (3)$$

Far away from the laser-induced hot spot, the temperature of the SC is equal to $T_0 < T_c$ and the order parameter $\psi_\infty$ is uniform, determined by $\psi_\infty^2 = \alpha(T_c - T_0)/\beta$. Substituting the dimensionless order parameter $\Psi = \psi/\psi_\infty$ into the

Ginzburg–Landau equation, we get

$$-\frac{\hbar^2}{4m}\frac{\partial^2\Psi}{\partial r^2} + \alpha[T(r) - T_c]\Psi + \alpha(T_c - T_0)\Psi^3 = 0 \qquad (4)$$

After division by $\alpha(T_c - T_0)$ and introduction of the superconducting coherence length $\xi_0$ defined by $\xi_0^2 = \frac{\hbar^2}{4m\alpha T_c}$, the equation writes

$$-\frac{\xi_0^2}{1 - T_0/T_c}\frac{\partial^2\Psi}{\partial r^2} + \frac{T(r) - T_c}{T_c - T_0}\Psi + \Psi^3 = 0, \qquad (5)$$

and is solved with the boundary conditions $\psi(r=0)=0$ and $\psi(r \to \infty)=1$.

**Data availability.** All relevant data are available from the corresponding author.

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

## Acknowledgements

We thank V.A. Skidanov and V.S. Stolyarov for providing the MO indicator and the niobium sample, respectively. We acknowledge the support by the European NanoSC COST Action MP1201, grants from the French National Agency for Research (ANR, project Electrovortex), Région Aquitaine, Idex Bordeaux (LAPHIA Program), the French Ministry of Education and Research, the Institut universitaire de France and the Fonds de Recherche du Québec—Nature et Technologies (fellowship for A.G.G.).

## Author contributions

B.L. introduced the concepts; P.T. and B.L. designed the experiments, which have been performed by I.S.V. and W.M.; A.G.G. and J.-B.T made the software for vortex manipulation; S.V.M. and A.I.B. performed the theoretical modelling; I.S.V., W.M., S.V.M., A.I.B., Ph.T. and B.L. analysed the results and wrote the manuscript; A.I.B., P.T. and B.L. supervised the project.

## Additional information

**Competing financial interests:** The authors declare no competing financial interests.

