## [Peer review file · Nature Communications]

Reviewers' Comments:

Reviewer #1 (Remarks to the Author)

Report on "Optical manipulation of single flux quanta" by Veshchunov et al.

The authors describe an elegant experiment demonstrating manipulation of a single vortex using a focused laser beam. They assert that this technique may be used for sculpting the magnetic flux profile in superconducting devices and may pave the way to fast optical drive of Josephson junctions.

The manuscript is well written, presenting data of high quality. Also, the theoretical analysis described in the "Methods" section is clear and complement the experimental work. I would recommend publication of this manuscript in Nature Communication provided the authors can better clarify the new physics emerging from their experiments. The vortex manipulation technique developed by the authors relies on the well-known physics of vortex flow by thermal gradients (Refs. 11, 12 of the present manuscript and reference therein). Their claim that "the performed experiments unveil new aspects of the interaction between laser radiation and the vortex matter" should be justified. The authors demonstrated the value of their technique in studying, e.g., the vortex behavior at the S/N boundary and the temperature dependence and the spatial distribution of the pinning force. Again, I would expect them to clarify what new physics and new insight emerged from these experiments.

Finally, Refs. 4-8 cited in the abstract of the manuscript are not the only publications describing manipulation of a single vortex. For example, the Oslo group has also described "manipulation of vortices by magnetic domain walls" and this work should also be cited. (This reference was copied from the Oslo group homepage; other relevant references - perhaps better ones - may be found therein).

Reviewer #2 (Remarks to the Author)

Veshchunov and colleagues demonstrate a method of optically depinning and manipulating the positions of magnetic flux vortices in type-II superconductors. The removal and manipulation of vortices in superconductors has been of interest for many years, and various methods have been demonstrated to manipulate vortices. The present authors add to the vortex manipulation toolbox by using a tightly focused laser beam to locally heat the SC sample, free nearby vortices from pinning sites, and move the vortices within the SC sample by moving the laser beam's position. The use of a focused laser beam, as well as the use of a thermal force, could open up new and simpler possibilities for vortex matter manipulation. The authors make a good case for the versatility, novelty, and usability of this method, and I lean towards recommending this manuscript for publication. The paper is also clear and straightforward, and should be accessible to a relatively wide audience. Prior to a recommendation, however, there are numerous details that I believe should be clarified or expanded upon in this paper. These are listed below.

1. The Gaussian radius r_0 mentioned in the methods is not used in the standard way for quoting Gaussian beam radii, unless the heat profile does not match the intensity profile (conventionally, the beam radius is the point at which intensity drops to $1/e^2$ of its peak value, so the 2 in the denominator of the heat profile Gaussian would be in the numerator). Given this, does the value of r_0 used in the calculation match up correctly with the FWHM of 1.1 microns quoted in the caption of Fig 1? Please clarify. Also: what is the laser used and its wavelength? How

were these parameters chosen? What effect would a smaller or larger beam radius have on vortex manipulation?

2. It would seem to me (perhaps naively) that a "vortex broom" might be more optimally created from a highly elongated Gaussian beam, with a 1D sweep through the sample. Has this been tried? Would it work?

3. What is the limit to the number of vortices that can be piled up in the center of a moving beam?

4. What are the time scales involved in this experiment? I would like to know: how long the beam needs to be on before the sample is sufficiently heated and re-thermalized (presumably this is nearly instantaneous?), how long the algorithm of vortex locating, depinning, and moving takes (per vortex, and for an area such as in figure 1c), how long a vortex broom sweep takes to perform (such as image 1e), and how long the vortex arrangement or vortex-free arrangement (fig 1 c or e) remains in place after the manipulation and sweep are performed. What limits the speed of this technique? How fast can a beam be moved before a vortex that is being manipulated by the beam is lost from the beam?

5. "Absorbed" beam power is quoted a few times. Presumably, the full laser power is 2.5 times this number, since 40% of the beam power is absorbed. Please clarify.

6. How efficient is the vortex depinning and manipulation technique? In other words: for a given vortex, what is the rate of success for placing the beam by the vortex and moving it to a desired location, all in the first attempt? For the vortices in Fig 1c, how many of them had to be "picked back up" by the laser beam before they were finally deposited at the desired locations? It's difficult to judge this from the movie provided, but since the final frame of the movie is different from the figure in the paper, some additional detail on efficiency would be good to provide.

7. How far can a vortex be dragged by the beam before it is lost from the beam (on average), at least for the samples studied? I assume that this distance is at least the width of the sample, since all vortices in a desired area can be removed, but I'm wondering if there are other limits on this manipulation technique that the authors explored that would be worthwhile mentioning.

8. If a vortex is positioned at a desired location, and then deposited there and held by pinning forces, how close can a second vortex be deposited to the first vortex, without the first vortex becoming depinned by the laser beam that is now positioning the second vortex?

Reviewer #3 (Remarks to the Author)

The manuscript of Veshchunov et al. presents an elegant and interesting new method for study and manipulation of individual vortices. The authors developed MO imaging with single vortex resolution, which has been attained in the past with only limited success. By adding a scanning focused laser beam the authors can readily move individual vortices over macroscopic distances and study their depinning due to the local heating. Although limited to quite low fields, this technique can become a useful tool for study of vortex pinning and dynamics and of the local effect of vortices on Josephson junctions. The manuscript is clearly written and the presented data and movies are vivid and informative. The paper could be suitable for publication in Nature Communications after the authors consider the following comments.

1. In the abstract the authors mention that they realize a "non-invasive manipulation" of vortices. This is not an appropriate terminology since heating the sample locally is clearly invasive and more generally any active manipulation is invasive.

2. The authors correctly point out in the abstract that handling of individual vortices has been

demonstrated only in few cases. They should note additional recent work of Embon et al., Scientific Reports 5, 7598 (2015) which also provides a quantitative measurement of the pinning force of an individual vortex comparable to the values concluded in the current study.

3. The quantitative evaluation of the pinning force is derived based on simulations of the induced temperature profiles and theoretical analysis of the resulting force on the vortex. These are valid calculations but clearly are parameter dependent and do not take into account additional nonequilibrium effects including Kapitza resistance. The derived numbers therefore can only be used as order of magnitude estimates and should be pointed out as such.

4. Similarly, the authors present the extended data Figure 3 as an evidence that the used laser power is sufficiently high to depin vortices but sufficiently low not to cause local heating to exceed T_c . I doubt that such a conclusion can be definitively reached from the presented data. Since the calculated diameter of the hot spot in extended Fig. 2 is comparable to the stated size of the vortex of $2.5 \mu\text{m}$ as observed by MO one cannot exclude the possibility that in extended Fig. 3 the peak temperature due to local heating exceeds T_c . In this case a normal region of μm scale diameter will be formed with a flux of Φ_0 trapped in it which will look like a vortex in MO imaging. The presented technique is still valid even if the peak temperature of the hot spot exceeds T_c but the quantitative analysis of the depinning force will be strongly affected. The numerical calculations can still be used to estimate the local temperature to be below T_c , but I do not think that the data presented in extended Fig. 3 can serve as an experimental proof of that but rather only as an upper bound on the diameter of the hot region with $T > T_c$.

5. Finally, the flux focusing surrounded by vortex free region presented in Fig. 3 is ascribed to BL surface barriers. The BL surface barrier is present at SC-insulator interface but is known theoretically and experimentally to be strongly suppressed at SC-normal interface, which is the case in the presented experiment. Moreover, the BL is a microscopic barrier on the scale of coherence length that will be additionally strongly suppressed by the smooth onset of the order parameter due to temperature gradient and due to thermal activation at local temperatures close to T_c . However, in thin samples in perpendicular field there is another barrier which is the geometrical barrier due to shielding currents (Zeldov PRL 73, 1428 (1994)). This is a robust macroscopic barrier which gives rise to extended vortex free regions near the sample edges (see for example, Segev PRB 83, 104520 (2011)). The authors should reevaluate their analysis in terms of geometrical barrier.

Response to reviewers:

We gratefully thank all the Reviewers for their thorough study of our manuscript and their valuable feedback, which have been very helpful for us to improve the presentation of our paper. We are delightful to see that they have found our experiment of broad interest. We have made revisions following their guidance. Below is an itemized reply to each point raised by the Reviewers.

Reviewer #1

We thank the referee for his extremely positive comments and his recommendation for publication. We address below his request for discussion and addition of citation.

1- *"I would recommend publication of this manuscript in Nature Communication provided the authors can better clarify the new physics emerging from their experiments. The vortex manipulation technique developed by the authors relies on the well-known physics of vortex flow by thermal gradients (Refs. 11, 12 of the present manuscript and reference therein). Their claim that "the performed experiments unveil new aspects of the interaction between laser radiation and the vortex matter" should be justified. The authors demonstrated the value of their technique in studying, e.g., the vortex behavior at the S/N boundary and the temperature dependence and the spatial distribution of the pinning force. Again, I would expect them to clarify what new physics and new insight emerged from these experiments."*

Response:

Our new method of single vortex manipulation addresses a new aspect of vortex movements in the presence of thermal gradients. The experiments are performed in the case of strong thermal gradients. The origin of the thermal force does not rely on the early observations of vortex flows in weak thermal gradients (Refs. 11, 12), where *"motion of fluxoids from the hot end to the cold end of the specimen"* (sentence taken from Ref. 11) was interpreted on the basis that vortices behave as entropy carrying particles. The sign of the force is opposite and its origin is introduced in the second paragraph of the manuscript.

In the revised version of the manuscript, we added the sentence *"This force will therefore drive vortices toward higher temperature regions"* at the end of the second paragraph in order to contrast with the sentence *"The early observations of vortex flows in weak thermal gradients were interpreted on the basis that vortices behave as entropy carrying particles, which obey general thermal diffusion laws and therefore are in search of colder places in the superconductor"* which is at the beginning of that paragraph.

The combination of laser illumination/switch off and magneto-optical imaging presented in this work allows, for the first time, the investigation of magnetic flux redistribution through moving N/S boundaries. The observed flux distribution is modeled by solving the Ginzburg Landau equation and physical image is given described in the frame of Bean-Livingston barrier.

Finally, our method can be used to estimate the magnitude of vortex pinning forces in our sample, which we found in agreement with earlier reports. This comforts the origin of the thermal force described in the second paragraph of the manuscript.

2- The reviewer also mentions that “Refs. 4-8 cited in the abstract of the manuscript are not the only publications describing manipulation of a single vortex. For example, the Oslo group has also described “manipulation of vortices by magnetic domain walls” and this work should also be cited”

Response:

We thank the referee for attracting our attention on this relevant work. The pioneering article of Goa, et al. entitled “*Manipulation of vortices by magnetic domain walls*” published in Applied Physics Letters 82, 79–81 (2003) is now cited in the first paragraph of the manuscript.

Reviewer #2

We thank the referee for his extremely positive comments and his recommendation for publication. Below we address the requested points of clarification.

1- “*The Gaussian radius r_0 mentioned in the methods is not used in the standard way for quoting Gaussian beam radii, unless the heat profile does not match the intensity profile (conventionally, the beam radius is the point at which intensity drops to $1/e^2$ of its peak value, so the 2 in the denominator of the heat profile Gaussian would be in the numerator). Given this, does the value of r_0 used in the calculation match up correctly with the FWHM of 1.1 microns quoted in the caption of Fig 1? Please clarify. Also: what is the laser used and its wavelength? How were these parameters chosen? What effect would a smaller or larger beam radius have on vortex manipulation? »*

Response:

Calculations were indeed performed with this conventional definition for a Gaussian distribution:

$$f(r) = \exp\left(-\frac{r^2}{2r_0^2}\right).$$

All FWHM values given in the manuscript were derived using the relation $\text{FWHM} = 2\sqrt{2 \ln 2} r_0$. Everything is consistent in the manuscript.

Any visible laser wavelength was suitable to single vortex manipulation technique, since the absorption spectrum of the superconductor is very broad. A laser with wavelength 561 nm (Coherent, continuous wave OBIS Laser) was used. We added in the legend of Figure 1 the wavelength of the laser. We also replaced “cw” by “continuous wave” for clarity.

The key parameter for single vortex manipulation method is the temperature gradient, which depends on the laser beam radius, the laser power and the position of the vortex with respect to the laser spot. At a given laser power, the maximal force is obtained for the tightest laser spot (diffraction limit) and at a vortex position corresponding to the maximal temperature gradient (see Supplementary Fig. 2).

Enlarging the beam radius will decrease the temperature gradient and therefore the thermal force. Keeping a constant thermal force while enlarging the beam radius will thus require a higher laser power for vortex manipulation, which may destroy superconductivity.

2- *"It would seem to me (perhaps naively) that a "vortex broom" might be more optimally created from a highly elongated Gaussian beam, with a 1D sweep through the sample. Has this been tried? Would it work? »*

Response:

In the present work we implemented a proof-of-principle vortex-broom with high efficiency. More refined optical designs will be investigated in future experiments in order to maximize the speed of the vortex-broom. A single 1 D sweep with a tailored elongated Gaussian beam, as suggested by the referee, will work. An elongated beam could indeed enable an extended broom width together with a strong temperature gradient in the transverse direction, and will therefore act as an efficient and larger "vortex-broom". These investigations are beyond the scope of the present manuscript.

3- *"What is the limit to the number of vortices that can be piled up in the center of a moving beam? »*

Response:

We did not explore yet this limit, which will be given by a balance between the attractive thermal force and the repulsive vortex-vortex force. In the experimental conditions of the vortex-broom manipulation displayed in Figure 1 e and the Supplementary movie 2, the number of vortices grabbed together by the moving laser beam along its path is estimated to 20.

4- *« What are the time scales involved in this experiment? I would like to know: how long the beam needs to be on before the sample is sufficiently heated and re-thermalized (presumably this is nearly instantaneous?), how long the algorithm of vortex locating, depinning, and moving takes (per vortex, and for an area such as in figure 1c), how long a vortex broom sweep takes to perform (such as image 1e), and how long the vortex arrangement or vortex-free arrangement (fig 1 c or e) remains in place after the manipulation and sweep are performed. What limits the speed of this technique? How fast can a beam be moved before a vortex that is being manipulated by the beam is lost from the beam? »*

Response:

- The time needed to induce a temperature profile strongly depends on the nature and thickness of the superconductor and its substrate. In the case of our niobium film (thickness 90 nm) grown on a silicon substrate (thickness 500 μm), we estimate a thermal response of the superconductor on a timescale of the order of 5 ns at 4.6 K.

- The algorithm of vortex localization, depinning and moving was run in 120 seconds for the supplemental video 1 and 100 seconds for Fig 1c. In the movie, we chose a low manipulation speed in order to image the movements of vortices with our magneto-optical imaging system with sufficient contrast. This required an integration time of 1.6 second per frame. The travel time per vortex trajectory was around 6 s in average corresponding to an average speed of $13 \mu\text{m/s}$.
- Concerning the vortex broom cleaning, the duration of a laser sweep is $\sim 13 \text{ s}$ ($11 \mu\text{m/s}$). The complete cleaning displayed in the Supplementary Movie 2 took 7 minutes. In all these experiments, we didn't aim at optimizing the vortex manipulation speed. We aimed at recording the vortex broom at work using our magneto-optical imaging system. In our setup the cleaning speed is limited by the laser scanning system (galvanometric mirrors and the driving electronics). An area of $70 \mu\text{m} \times 70 \mu\text{m}$ could be cleaned in 350 ms (corresponding to $\sim 7 \text{ mm/s}$ for the vortex drive speed). Faster scanners would even increase the cleaning speed.
- The maximal driving speed of a vortex (i.e. the speed above which a vortex is lost by the scanned laser beam) should be given by the ratio of the hotspot size to the thermal response time of the sample ($1 \mu\text{m/ns}$ is an order of magnitude).
- After the laser switch off, the vortices remain trapped by local pinning sites. The vortex-arrangements and vortex free arrangements are therefore permanent.

We added in the discussion section of the manuscript these two small paragraphs :

"By choosing the appropriate laser parameters, we showed that one can perform a safe and efficient manipulation of single vortices, with a 100% rate of success. In practice, the distance over which a vortex can be dragged is only limited by the field of view of the microscope objective, which is of the order of a millimeter in our case. ...

In the present work we implemented a proof-of-principle vortex-broom and aimed at recording the vortex broom at work, using our magneto-optical imaging system. In the supplemental video 2, we chose a low manipulation speed in order to image the movements of vortices with a sufficient contrast. The complete cleaning displayed in this video took 420 s, corresponding to an average vortex speed of $10 \mu\text{m.s}^{-1}$. Actually, the laser scanning system, including galvanometric mirrors and the driving electronics, limits the cleaning speed in our setup. An area of $70 \mu\text{m} \times 70 \mu\text{m}$ could be cleaned in 350 ms, corresponding to $\sim 6 \text{ mm.s}^{-1}$ for the vortex drive speed. Faster scanners would increase the vortex driving speed, up to a maximal speed which should be given by the ratio of the hotspot size ($\sim 1 \mu\text{m}$) to the thermal response time of the sample ($\sim 1 \text{ ns}$), of the order of 1 km.s^{-1} ."

5- "Absorbed" beam power is quoted a few times. Presumably, the full laser power is 2.5 times this number, since 40% of the beam power is absorbed. Please clarify".

Response:

The power of the laser beam impinging on the superconductor is indeed 2.5 times the "absorbed" power. We chose to refer to the absorbed power rather than the laser power at the entrance of the cryostat for more relevance, since it takes into account the transmission losses in our optical setup.

6- « How efficient is the vortex depinning and manipulation technique? In other words: for a given vortex, what is the rate of success for placing the beam by the vortex and moving it

to a desired location, all in the first attempt? For the vortices in Fig 1c, how many of them had to be "picked back up" by the laser beam before they were finally deposited at the desired locations? It's difficult to judge this from the movie provided, but since the final frame of the movie is different from the figure in the paper, some additional detail on efficiency would be good to provide. »

Response:

In the region of the superconductor displayed in Fig. 2a, we demonstrate in Figure 2c that the rate of success for moving any vortex with a laser beam is 100% if the laser power is high enough. For three different sample temperatures, we show that such a power can be found. We also checked in other regions that 100 % of the vortices could be dragged out of their pinning sites.

In Fig 1c and in the video 1, we did not pick back any vortex. Manipulating single vortices across a region containing other vortices that one wants to maintain requires very fine tuning of the laser power, so that all vortices chosen for manipulation are untrapped with a maximal success rate, while avoiding grabbing other vortices on the laser beam path. In Fig. 1c, the laser was well adjusted and no vortex was picked back up. In the supplementary video 1, a vortex is missing in the letter A probably because the power was not high enough to untrap all the vortices in this region.

We added in the discussion section of the manuscript *"By choosing the appropriate laser parameters, we showed that one can perform a safe and efficient manipulation of single vortices, with a 100% rate of success."*

7- *« How far can a vortex be dragged by the beam before it is lost from the beam (on average), at least for the samples studied? I assume that this distance is at least the width of the sample, since all vortices in a desired area can be removed, but I'm wondering if there are other limits on this manipulation technique that the authors explored that would be worthwhile mentioning. »*

Response:

In practice, the distance over which a vortex can be dragged will be limited by the field of view of the microscope objective, which is typically of the order of a millimeter in our case. Note that from the videos we can directly see that a vortex can be driven over distances larger than a hundred microns.

We added in the discussion section of the manuscript: *"In practice, the distance over which a vortex can be dragged is only limited by the field of view of the microscope objective, which is of the order of a millimeter in our case."*

8- *« If a vortex is positioned at a desired location, and then deposited there and held by pinning forces, how close can a second vortex be deposited to the first vortex, without the first vortex becoming depinned by the laser beam that is now positioning the second vortex? »*

Response:

Since vortices are imaged with a magneto-optical technique whose resolution is about 2.5 μm in our experimental conditions (see Supplementary information), it is

difficult to investigate the minimal distance between two vortices. This distance is set likely to be set by the vortex-vortex repulsion and would be of the order of the London penetration depth ($\lambda \sim 100$ nm for a Niobium sample).

If the laser spot drags a vortex to the vicinity of another vortex (within the vortex localization accuracy), we expect that after the laser switch-off the incoming vortex will be released to the nearest pinning sites. The initial vortex might be untrapped by the thermal force, and attracted towards the incoming vortex. Such a scenario could be investigated in an experiment which combines optical manipulation and local probe scanning microscopy technique (such as MFM).

Reviewer #3

We thank the referee for his extremely positive comments and his recommendation for publication. Below we address the requested points of clarification.

1- *« In the abstract the authors mention that they realize a "non-invasive manipulation" of vortices. This is not an appropriate terminology since heating the sample locally is clearly invasive and more generally any active manipulation is invasive. »*

Response:

We agree with the referee and removed « *non-invasive* » in the revised version of the manuscript.

2- *« The authors correctly point out in the abstract that handling of individual vortices has been demonstrated only in few cases. They should note additional recent work of Embon et al., Scientific Reports 5, 7598 (2015) which also provides a quantitative measurement of the pinning force of an individual vortex comparable to the values concluded in the current study. »*

Response:

We thank the referee for bringing this work to our attention. We have now included this reference in the introductory paragraph of the revised manuscript.

3- *« The quantitative evaluation of the pinning force is derived based on simulations of the induced temperature profiles and theoretical analysis of the resulting force on the vortex. These are valid calculations but clearly are parameter dependent and do not take into account additional nonequilibrium effects including Kapitza resistance. The derived numbers therefore can only be used as order of magnitude estimates and should be pointed out as such ».*

Response:

We fully agree with the referee. In the revised version of the manuscript, we replaced on page 4 « *straightforward determination of the pinning force* » by

« straightforward estimation of the pinning force ». We also replaced « this value » by « this order of magnitude estimate » a few sentences later in the text.

4- « Similarly, the authors present the extended data Figure 3 as an evidence that the used laser power is sufficiently high to depin vortices but sufficiently low not to cause local heating to exceed T_c . I doubt that such a conclusion can be definitively reached from the presented data. Since the calculated diameter of the hot spot in extended Fig. 2 is comparable to the stated size of the vortex of $2.5 \mu\text{m}$ as observed by MO one cannot exclude the possibility that in extended Fig. 3 the peak temperature due to local heating exceeds T_c . In this case a normal region of μm scale diameter will be formed with a flux of Φ_0 trapped in it which will look like a vortex in MO imaging. The presented technique is still valid even if the peak temperature of the hot spot exceeds T_c but the quantitative analysis of the depinning force will be strongly affected. The numerical calculations can still be used to estimate the local temperature to be below T_c , but I do not think that the data presented in extended Fig. 3 can serve as an experimental proof of that but rather only as an upper bound on the diameter of the hot region with $T > T_c$. »

Response:

We thank the referee for this remark, which gives us the opportunity to put more emphasis on the fact that *the used laser power are sufficiently low to avoid local heating above T_c .*

To prove this, we have compared the magneto-optical images of vortices under laser illumination (Supplementary Fig. 3) with that of vortices at different SC temperatures (without laser, Supplementary Fig. 1). Since the vortex image profile (width, contrast) is slightly affected by the laser, we conclude that the local temperature remains well below T_c when performing the vortex manipulation.

We added in the revised version of the manuscript: *“For this, we have compared the magneto-optical images of vortices under laser illumination (see the Supplementary Fig. 3) with that of vortices at different SC temperatures (without laser, see the Supplementary Fig. 1). Since the vortex image profile (width, contrast) is slightly affected by the laser, we conclude that the local temperature remains well below T_c when performing the vortex manipulation.”*

5- « Finally, the flux focusing surrounded by vortex free region presented in Fig. 3 is ascribed to BL surface barriers. The BL surface barrier is present at SC-insulator interface but is known theoretically and experimentally to be strongly suppressed at SC-normal interface, which is the case in the presented experiment. Moreover, the BL is a microscopic barrier on the scale of coherence length that will be additionally strongly suppressed by the smooth onset of the order parameter due to temperature gradient and due to thermal activation at local temperatures close to T_c . However, in thin samples in perpendicular field there is another barrier which is the geometrical barrier due to shielding currents (Zeldov PRL 73, 1428 (1994)). This is a robust macroscopic barrier which gives rise to extended vortex free regions near the sample edges (see for example, Segev PRB 83, 104520 (2011)). The authors should reevaluate their analysis in terms of geometrical barrier. »

Response:

We warmly thank the Reviewer for attracting our attention on this important point. We agree with the reviewer that the geometrical barrier is more relevant than the Bean-Livingston barrier in our experimental conditions. Following the theoretical line of the seminal work of Zeldov et al. (PRL 73, 1428, 1994), we derived a model of geometrical barrier which reproduces very well the experimental data of Fig. 3f without fitting parameter. The details of this model are given in the revised version of the Supplementary information file. We also replaced “Bean-Livingston barrier” by “geometrical barrier” and added two references (Zeldov et al. PRL 73, 1428 (1994); Segev et al. PRB 83, 104520 (2011)) in the revised manuscript.

Reviewers' Comments:

Reviewer #1 (Remarks to the Author)

The authors responded to my criticism and comments.
I recommend publishing the revised version.

Reviewer #2 (Remarks to the Author)

The authors have suitably addressed all referee comments and questions in the revised manuscript. I have no further questions for the authors, and am happy to recommend this manuscript for publication in Nature Communications.

Reviewer #3 (Remarks to the Author)

The authors have addressed the comments of the referees. I recommend publication of the revised manuscript.